# The role of personality in neighborhood satisfaction

Zachary P. Neal *, Brian Brutzman

Psychology Department, Michigan State University, East Lansing, MI, United Stated of America

* zpneal@msu.edu

## Abstract

Urbanists have long been interested in understanding what makes people satisfied with their neighborhoods. However, relatively little is known about how residents' personality traits may affect their neighborhood satisfaction. In this paper, we explore the direct and indirect associations of personality traits with neighborhood satisfaction in a representative sample of adults in Michigan (USA). We find that each of the personality traits in the five factor model are associated with neighborhood satisfaction in the same way that they are known to be associated with subjective well-being. However, we fail to observe evidence that personality traits moderate the association between perceptions of neighborhoods and neighborhood satisfaction, or that personality's association with neighborhood satisfaction is mediated by neighborhood perceptions. We conclude that there is potential for drawing on theoretical and empirical developments in positive psychology for understanding neighborhood satisfaction, but observe that the underlying mechanisms for the association between personality and neighborhood satisfaction remain unknown.

## Introduction

Urbanists have long been interested in understanding what makes people satisfied with their neighborhoods, with different theoretical models pointing to different explanatory variables [1, 2]. Although past research has considered a wide range of potential explanations for variation in neighborhood satisfaction, with mixed results, relatively little is known about how residents' psychological characteristics may affect their neighborhood satisfaction. The majority of research on psychological characteristics and neighborhood satisfaction has focused on the role of subjective well-being [3–6]. However, the role of personality, which can be readily measured using validated scales and is known to be associated with other dimensions of well-being, remains largely unexplored. This is surprising because for over a century urban theorists have speculated that personality plays a role in how cities are experienced. For example, early theorists wrote that cities "are the genuine locale of the blasé attitude" [7], and about "temperament and the urban environment" [8]. More recently, urban planners have speculated that personality may be important for planning 'happy neighborhoods,' calling for more research on the links between personality and neighborhoods [9]. Therefore, in this paper we investigate the direct and indirect roles of personality in neighborhood satisfaction.

**Data Availability Statement:** All data and materials necessary to reproduce the findings are available at https://osf.io/wguds.

**Funding:** The author(s) received no specific funding for this work.

**Competing interests:** The authors have declared that no competing interests exist.

Personality is often conceptualized as consisting of five traits: openness to experience, conscientiousness, extraversion, agreeableness, and neuroticism [10]. We investigate how these personality traits are associated with neighborhood satisfaction in a representative sample of adults in Michigan (USA). Consistent with prior research on the association between subjective well-being and personality traits [11–16], we find that neighborhood satisfaction is significantly positively correlated with conscientiousness, extraversion, and agreeableness, and significantly negatively correlated with neuroticism. We also find that openness is not significantly correlated with neighborhood satisfaction, which is consistent with prior research on the weak and contingent (e.g., only for older adults) association between openness and life satisfaction [17].

While personality traits may have a direct association with neighborhood satisfaction, prior theory suggests they could also have indirect effects. The person-environment fit model that we develop here suggests that personality moderates the relationship between perceived neighborhood characteristics and neighborhood satisfaction, such that individuals with different personalities derive satisfaction from different neighborhood characteristics. Similarly, the rose-colored glasses model that we develop here suggests that personality's effect on neighborhood satisfaction is mediated by neighborhood perceptions, such that individuals with different personalities perceive their neighborhoods in different ways, which in turn accounts for differing levels of satisfaction. However, focusing on the perceived neighborhood characteristics of social cohesion and disorder, we find no support for these theoretically-motivated models of indirect associations. Thus, although personality is associated with neighborhood satisfaction, the mechanisms of that association remain unclear.

The paper is organized in four sections. In the background section, we develop an integrated model of neighborhood satisfaction, then review the prior literature on the direct and indirect roles of personality. In the methods section, we describe the State of the State data, and our analytic strategy for exploring the association between personality and neighborhood satisfaction. In the results section, we report the direct association between personality traits and neighborhood satisfaction, as well as their indirect moderating and mediated effects on neighborhood satisfaction. Finally, in the discussion section, we summarize our findings, noting the study's limitations, and speculating on alternative mechanisms for the observed association between personality and neighborhood satisfaction.

## Background

### Pathways to neighborhood satisfaction

Several theoretical models have been proposed for explaining the factors associated with neighborhood satisfaction. Before turning to the role of personality, we first describe an integrated perspective on these multiple pathways to neighborhood satisfaction (see Fig 1).

The *compositional* [18] or *systemic* [19] model focuses on individual differences as an explanation for variation in neighborhood satisfaction (Path A in Fig 1). Studies investigating this model typically focus on demographic characteristics, aiming to identify those that are associated with neighborhood satisfaction. For some characteristics, the association is relatively robust; for example, women and older adults tend to be more satisfied with their neighborhoods [20–22]. However, for other characteristics such as educational attainment, the association has been mixed, with studies finding it is positively [20], negatively [21], or not [22] associated with neighborhood satisfaction.

The *subjective* model [18] focuses on individuals' perceptions of neighborhood characteristics (Path B in Fig 1). Studies investigating this model ask residents about their perception of specific neighborhood characteristics (e.g., what do you think of the schools?), then aim to

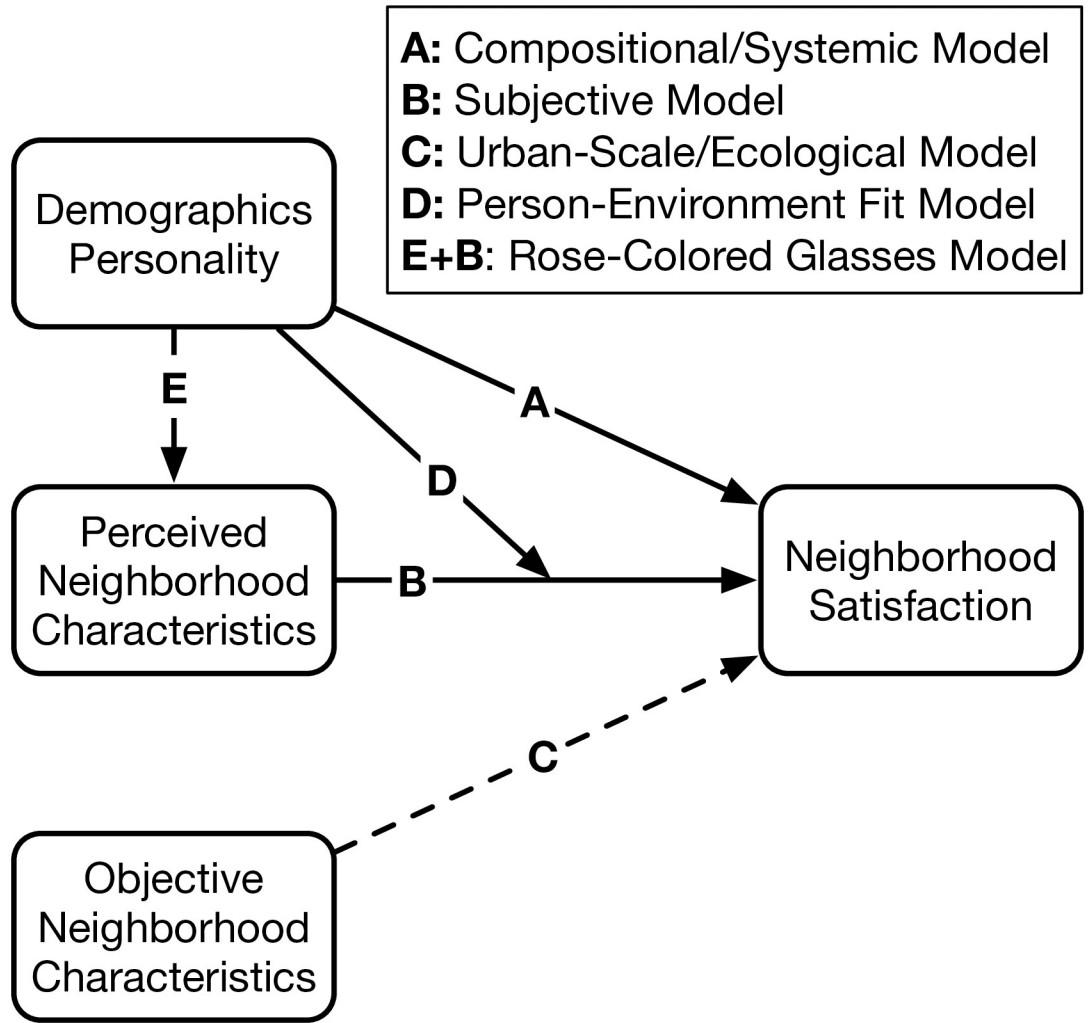

**Fig 1. Pathways to neighborhood satisfaction.**

identify those characteristics for which the resident's perception is associated with their neighborhood satisfaction. Importantly, this model focuses on subjective perceptions (e.g., perceived quality of schools), not objective reality (e.g., actual school performance). Accordingly, when it comes to characteristics that contribute to individuals' neighborhood satisfaction, "discrepancies between reality and perception often occur" [1]. While there is some variation in the perceived neighborhood characteristics that studies have found is associated with neighborhood satisfaction, there is also substantial agreement on some relatively intuitive characteristics. For example, studies routinely find that residents are more satisfied with neighborhoods they perceive to be socially cohesive [23, 24] (i.e., characterized by social bonds among the residents [25–30]), and less satisfied with neighborhoods they perceive to exhibit disorder [31] (i.e., characterized by crime, trash, and vacancy [22, 25, 30, 32, 33]). Because social cohesion and neighborhood disorder are among the most consistently associated with neighborhood satisfaction, we focus on these two perceived neighborhood characteristics in our analyses below.

The *urban scale* [18] or *ecological* [19] model focuses on objective neighborhood characteristics (Path C in Fig 1). Some studies adopting this perspective rely on official statistics

(e.g., crime rates, school test scores) or direct measurements (e.g., square meters of parkland) to identify this association. Other studies rely on statistical indices (e.g., intraclass correlation coefficient, Moran's $I$) to detect spatial clustering of neighborhood satisfaction that would be expected if objective characteristics played a role. Despite the intuitive appeal of this model, which is the foundation of neighborhood planning-focused efforts to build neighborhoods people will find satisfying, it remains generally unsupported. For example, a large study of residents in Seattle and Baltimore found that "Seven of nine perceived environment characteristics evaluated in the present study were significantly associated with neighborhood satisfaction, but none of the objectively assessed characteristics were significant" [29]. Similarly, a study of residents in Flint, Michigan found that "census variables [capturing objective neighborhood characteristics] did not add information to a model of neighborhood satisfaction already containing survey variables [capturing residents' perceptions of their neighborhoods]" [34]. Pooling these findings, a recent meta-analysis found that only about 15% of the variation in neighborhood satisfaction could be explained by characteristics of the neighborhoods themselves [2]. Because the urban scale/ecological model has received limited empirical support and does not involve the role of personality, we do not investigate this model here, but for the sake of completeness include it (with a dashed line) in Fig 1.

These models hypothesize direct effects of individual and neighborhood characteristics on neighborhood satisfaction. However, two additional models hypothesize more complex indirect effects. The *person-environment fit* model hypothesizes that neighborhood characteristics enhance neighborhood satisfaction for some groups, but not others, and thus that individual characteristics *moderate* the association between perceptions and satisfaction (Path D in Fig 1). Initial descriptions of this model noted that neighborhood satisfaction depends on "the fit between the [neighborhood], as perceived by the individual, and the standard against which the individual measured that perception," which can depend on the individual's needs and values [1], and has since been elaborated by others [35, 36]. In principle, this model hypothesizes that the relevant individual characteristics are preferences, needs, and values. For example, a person who enjoys parks and lives in a neighborhood where they perceive parks to be accessible will be satisfied, while in contrast a person who hates parks and lives in a neighborhood where they perceive parks to be accessible will be dissatisfied. However, in practice, studies investigating this model tend to focus on demographic characteristics as a proxy for preferences. For example, one study found that compared to couples without children, couples with children were more satisfied when they lived in neighborhoods with many other families with children [36]. Similarly, another study found that nightlife was significantly associated with neighborhood satisfaction for non-parents, but not for parents [20].

Finally, the *rose-colored glasses* model hypothesizes that individual characteristics affect how individuals perceive their neighborhood, which in turn affects their neighborhood satisfaction, and thus that the effect of individual characteristics on neighborhood satisfaction are *mediated* by perceptions (paths E & B in Fig 1). When considering how subjective perceptions of neighborhoods contribute to neighborhood satisfaction "the characteristics of the person must be considered" because discrepancies between reality and perception "can be investigated in relation to traits of the perceiver" [1]. For example, some individuals (colloquially, those who wear rose-colored glasses) may tend to perceive aspects of their neighborhood as better than they actually are, and these positive perceptions increase their neighborhood satisfaction. Despite this model's conceptual simplicity, we are not aware of any studies of neighborhood satisfaction that have directly tested it.

## The direct role of personality

Although many of these models of neighborhood satisfaction have considered the role of individual characteristics, they have generally focused only on demographic characteristics, while neglecting psychological characteristics such as personality. However, personality traits may also play an important role in neighborhood satisfaction.

The Five Factor Model is a dominant paradigm of assessment and research in personality psychology, enjoying robust empirical support [37–41]. The five factors or 'traits' specified by this model include openness to experience, conscientiousness, extraversion, agreeableness, and neuroticism. Each of these traits is associated with familiar patterns of behavior. People with openness to experience seek out new experiences, and are characterized by having both artistic and scientific creativity. Conscientious people typically present as competent, orderly, achievement oriented, and disciplined. Extraverts are the proverbial "life of the party," characterized by their warmth, energy, gregariousness, and assertiveness. Agreeable people tend to be compliant, cooperative, altruistic, and modest. Finally, people high in neuroticism tend to be anxious, hostile, depressed, and less self-assured.

Meta-analyses have consistently demonstrated the same pattern of association between these traits and satisfaction with life (SWL) [12, 42], satisfaction with job (SWL) [43], and subjective well-being (SWB) [11]. As Table 1 highlights, openness is weakly positively associated with these constructs; conscientiousness, extraversion, and agreeableness are moderately positively correlated with well-being; and neuroticism is moderately negatively correlated with well-being. These prior findings are informative for thinking about the potential role of personality in neighborhood satisfaction because neighborhood satisfaction is often viewed as a component or predictor of well-being or of satisfaction in other domains [3–6, 45, 46].

While this prior research on personality and well-being offers some insight, one study has directly examined the association between personality and indicators of neighborhood satisfaction. In a large ($N = 3,760$) sample of adults in the United States in 1995, this study examined the association between personality traits and two indicators of neighborhood satisfaction: neighborhood quality (e.g., "People in my neighborhood trust each other") and neighborhood equality (e.g., "Most people live in a better neighborhood than I do") [44]. As Table 1 highlights, the associations with both variables were similar in magnitude and direction to prior meta-analytic findings.

Drawing on meta-analytic findings that personality is associated with psychological constructs [11, 12, 42, 43] and early findings that personality is associated with neighborhood satisfaction [44], we ask: *Is personality associated with neighborhood satisfaction* (**Research Question 1**)? More specifically, we hypothesize that neighborhood satisfaction is weakly

**Table 1. Prior correlations between personality traits and psychological/neighborhood constructs.**

|  | Meta-Analyses of Psychological Constructs | | | | Neighborhood Quality [44] | Neighborhood Equality[a] [44] |
|---|---|---|---|---|---|---|
|  | **SWB [11]** | **SWL [12]** | **SWL [42]** | **SWJ [43]** |  |  |
| Openness | 0.11 | 0.03 | 0.08 | 0.04 | 0.12 | 0.14 |
| Conscientiousness | 0.21 | 0.22 | 0.27 | 0.20 | 0.17 | 0.26 |
| Extraversion | 0.17 | 0.28 | 0.32 | 0.23 | 0.19 | 0.20 |
| Agreeableness | 0.17 | 0.14 | 0.20 | 0.16 | 0.10 | 0.14 |
| Neuroticism | -0.22 | -0.38 | -0.39 | -0.26 | -0.20 | -0.21 |

SWB = Subjective well-being, SWL = Satisfaction with life, SWJ = Satisfaction with job

[a] The original study measured neighborhood *inequality*. Here, we reverse the measure and scores for comparability with other measures

positively associated with openness (**H1a**), moderately positively associated with conscientiousness (**H1b**), extraversion (**H1c**), and agreeableness (**H1d**), and negatively associated with neuroticism (**H1e**).

## The indirect roles of personality

We argue in the preceding section that personality may have a direct association with neighborhood satisfaction. However, in the broader literature on personality and subjective well-being there is also evidence to suggest more complex relationships between personality traits, other socio-cognitive constructs (i.e., values, attitudes, and beliefs), and well-being, including that such constructs might mediate or moderate the relationship between personality traits and subjective well-being [47]. Related longitudinal research points to a dynamic interplay of personality traits, values, and well-being over time which suggests that indirect relationships between these constructs bear further exploration [15]. In this section, we sketch two ways that personality may also have indirect associations with neighborhood satisfaction.

**Person(ality)-environment fit.**   Most studies of person-environment fit focus on the preferences or demographic characteristics of the 'person' [48], however personality also matters. Congruence models of personality hold that compatibility between an individual's personality and their environment bolsters subjective well-being. For example, "an extravert is happier in social situations. . .[and] if she lived in a cloistered nunnery. . .might be less happy" [49]. The things an extravert values in the environment (e.g., social engagement) differ from the things an introvert values (e.g., solitude).

One study observed evidence that subjective well-being is also bolstered by personality congruence specifically with neighborhood characteristics [50]. Examining a large sample of London (UK) residents ($N = 56,019$) in 216 postal districts, they found that the relationship between selected neighborhood characteristics and life satisfaction was moderated by three personality traits: openness, conscientiousness, and agreeableness. For example, they observed that "Living in a densely populated, ethnically heterogeneous neighborhood with low proportion of older people and families with children provided the best match for individuals with high openness to experience" [50]. Similarly, agreeableness was associated with higher life satisfaction in neighborhoods with children, while conscientiousness was associated with lower life satisfaction in neighborhoods with fewer seniors.

To our knowledge there is no prior research on personality congruence boosting neighborhood satisfaction, so we proceed in an exploratory fashion by asking: *Do personality traits moderate the association of neighborhood satisfaction with perceptions of neighborhood cohesion and disorder* (**Research Question 2**; path D in Fig 1)? To the extent that conscientious individuals are oriented toward order and discipline, we might expect a stronger negative association between neighborhood disorder and neighborhood satisfaction among highly conscientious individuals. Similarly, for the same reasons that ""an extravert is happier in social situations" [49], we might expect an extravert to be more satisfied in a socially cohesive neighborhood, and thus might expect a stronger positive association between neighborhood social cohesion and neighborhood satisfaction among highly extraverted individuals.

**Personality and rose-colored glasses.**   Personality psychologists have described many ways that personality shapes our interactions with the world, including *reactive transactional mechanisms* in which personality shapes how we perceive or interpret what we observe [51, 52]. People with different personalities often "react to similar environments in idiosyncratic but consistent ways," as when "an insecure person. . .[interprets] someone's actions as being threatening, regardless of the actual intentions of the actor" [51].

Only one study has examined this reactive mechanism in the context of perceptions of neighborhoods ([53], path E in Fig 1). They observed among older adults that all five personality traits were weakly associated with perceptions of neighborhood crime and cohesion. However, personality traits have consistently been associated with perceptions of crime and cohesion more broadly. First, the closely related personality traits of neuroticism (from the five-factor model) and emotionality [54] have been linked with greater fear of crime in online samples [55], samples of students and neighborhood residents [56], and samples of long-time neighborhood residents [57]. Second, both extraversion and agreeableness have been linked to perceptions of social cohesion among students [58], on team tasks [59], and in families [60].

Both crime and cohesion are well-known predictors of neighborhood satisfaction, and these findings shed some light on individual factors that influence perceptions of crime and cohesion. Because there is no prior research on reactive transactional mechanisms of personality affecting neighborhood satisfaction, we proceed in an exploratory fashion by asking: *Are personality traits' association with neighborhood satisfaction mediated by perceptions of neighborhood cohesion and disorder* (**Research Question 3**; paths E & B in Fig 1)? As an exploratory analysis, we consider all five personality traits. However, guided by past findings we might expect that the effect of neuroticism will be mediated by perceptions of disorder, while the effects of extraversion and agreeableness will be mediated by perceptions of cohesion.

## Methods

### Data

To examine the role of personality in neighborhood satisfaction, we analyze data from Michigan State University's State of the State Survey (SOSS). The SOSS is collected several times annually by the Institute for Public Policy and Social Research (IPPSR). Each SOSS survey consists of core items (e.g. demographic characteristics) and wave-specific items commissioned by researchers. In this wave, the items relating to neighborhood satisfaction and personality were commissioned by the first author, while the items relating to perceived neighborhood characteristics were commissioned by an unknown researcher not associated with this study. These data were collected between May 8 and 25, 2020 from a sample of 1000 Michigan adults, for which we use sampling weights to ensure population representativeness with respect to sex, age, race, and education.

Two features of the data collection timing are notable. First, the data were collected during the first wave of COVID-19 in Michigan, at a time when residents were under a stay-at-home order. Second, data collection concluded on the day that then-officer Derek Chauvin murdered George Floyd in Minneapolis, sparking widespread protests against police brutality. Therefore, these data offer insight into neighborhood satisfaction in the COVID-era, but are not influenced by the summer of Black Lives Matter (BLM) and related protests [61–63].

### Measures

**Neighborhood satisfaction.** Neighborhood satisfaction is measured by asking respondents: "*Taking everything into account, how satisfied are you with your neighborhood as a place to live?*" Responses were provided on a 5-point ordinal scale ranging from 'very satisfied' (5) to 'very dissatisfied' (1). This approach follows the widely-used convention of measuring neighborhood satisfaction with a single-item, with the number of response options in the middle of past studies that have used 10 [28], 7 [64], 5 [20], 4 [65], or 2 [66] options.

**Personality.** To measure personality traits, we used the 20-item Mini International Personality Item Pool [67]. The mini-IPIP includes five 4-item subscales for each of the Big Five personality traits (i.e., openness, conscientiousness, extraversion, agreeableness, and neuroticism).

Openness is measured with items such as *"I have a vivid imagination"*. Conscientiousness is measured with items such as *"I get chores done right away"*. Extraversion is measured with items such as *"I am the life of the party"*. Agreeableness is measured with items such as *"I sympathize with others' feelings"*. Finally, neuroticism is measured with items such as *"I have frequent mood swings"*. Each item was rated on a 5-point scale ranging from 'very inaccurate' (1) to 'very accurate' (5); subscale scores were computed by averaging across the four items associated with each trait, reverse coding when necessary. Internal consistencies were acceptable in the current sample (openness $\alpha = 0.71$, conscientiousness $\alpha = 0.66$, extraversion $\alpha = 0.78$, agreeableness $\alpha = 0.73$, neuroticism $\alpha = 0.72$).

**Perceived neighborhood characteristics.** Perceptions of neighborhood social cohesion were measured by asking respondents whether they agree with the statement "*I live in a close-knit neighborhood*", with response options ranging from 'strongly disagree' (1) to 'strongly agree' (5). Perceptions of neighborhood disorder were measured using the mean of respondents' agreement with two statements—"*Crime is a big problem in my neighborhood*" and "*Properties that are not looked after are a big problem in my neighborhood*"—with the same response options.

## Analysis plan

To examine the association between personality traits and neighborhood satisfaction, we compute the corresponding Pearson correlation coefficients. To examine whether personality moderates the relationship between neighborhood perceptions and neighborhood satisfaction, as hypothesized by the person-environment fit model, we estimate a series of linear regressions predicting neighborhood satisfaction as a function of one personality trait, one neighborhood perception, and their interaction. Finally, to examine whether personality's association with neighborhood satisfaction is mediated by neighborhood perceptions, as hypothesized by the rose-colored glasses model, we separately estimate the average causal mediated effect (ACME) of each personality trait through each neighborhood perception, using the `mediation` package for R [68]. This estimation technique is recommended for testing mediated effects, however despite the name of the ACME estimator, we do not aim to draw causal conclusions from these models because our data are cross-sectional and do not involve experimental manipulation. All analyses use the `survey` package for R [69] to obtain population representative estimates with survey weights, and all independent variables were mean centered prior to each model estimation. All materials necessary to reproduce the reported analyses are available at https://osf.io/wguds, which also include sensitivity tests (1) comparing the use of Spearman and Pearson correlation coefficients with ordinal neighborhood satisfaction data, and (2) the impact of potential violations of the sequential ignorability assumption on ACME estimates.

## Results

Table 2 reports the population estimates for neighborhood satisfaction, each personality trait, and our two focal perceived neighborhood characteristics, cohesion and disorder. As expected, we observe that the estimated population mean for most variables is near the midpoint of their five-point scales. However, consistent with prior literature on neighborhood satisfaction, we also observe that on average individuals tend to be highly satisfied with their neighborhoods ($M = 3.909$, $SE = 0.049$; [2]).

The bivariate correlations reported in column (1) of 2 provide an answer to Research Question 1: *is personality associated with neighborhood satisfaction?* Our hypotheses are generally supported by these results. First, we find that conscientiousness ($r = 0.162$, $p < 0.001$), extraversion ($r = 0.1$, $p = 0.003$), and agreeableness ($r = 0.217$, $p < 0.001$) are positively associated

**Table 2. Population estimates.**

|  | Mean | SE | Pearson correlation | | | | | | |
|---|---|---|---|---|---|---|---|---|---|
|  |  |  | (1) | (2) | (3) | (4) | (5) | (6) | (7) |
| (1) N'hood Satisfaction | 3.909 | 0.049 |  |  |  |  |  |  |  |
| (2) Openness | 3.638 | 0.037 | 0.037 |  |  |  |  |  |  |
| (3) Conscientious | 3.625 | 0.036 | 0.162 | 0.029 |  |  |  |  |  |
| (4) Extraversion | 2.743 | 0.040 | 0.100 | 0.124 | 0.134 |  |  |  |  |
| (5) Agreeableness | 3.892 | 0.036 | 0.217 | 0.250 | 0.209 | 0.275 |  |  |  |
| (6) Neuroticism | 2.744 | 0.038 | -0.243 | -0.073 | -0.242 | -0.168 | -0.086 |  |  |
| (7) Perceived Cohesion | 2.909 | 0.052 | 0.307 | -0.071 | 0.135 | 0.239 | 0.148 | -0.135 |  |
| (8) Perceived Disorder | 2.217 | 0.056 | -0.464 | -0.163 | -0.051 | -0.017 | -0.145 | 0.172 | -0.087 |

with neighborhood satisfaction, consistent with hypotheses H1b—H1d. Second, we find that neuroticism ($r = -0.243$, $p < 0.001$) is negatively associated with neighborhood satisfaction, consistent with hypothesis H1e. Finally, contrary to hypothesis H1a, we find that openness ($r = 0.037$, $p = 0.223$) is not significantly associated with neighborhood satisfaction, however we had hypothesized this association would be weak.

Table 3 reports a series of linear regressions predicting neighborhood satisfaction as a function of a personality trait, a perceived neighborhood characteristic, and their interaction, and thus provide an answer to Research Question 2: *does personality moderate the association between perceptions and satisfaction?* In this table, each row reports the results of one model. For example, the 'IV = Disorder, M = Openness' model reported in the first row reports the results of a model in which the association between perceived neighborhood disorder and neighborhood satisfaction is moderated by openness. Within each model, we observe that neighborhood perceptions always have a statistically significant direct association with neighborhood satisfaction. As expected from the correlations shown in Table 1, we also observe that most personality traits have a statistically significant direct association with neighborhood satisfaction. However, as shown in the final column, none of the interaction terms in these models are statistically significant. This indicates that personality does not moderate the association between perceptions and satisfaction, and fails to support the *person-environment fit* model (path D in Fig 1) with respect to personality.

**Table 3. Personality traits as moderators (M) of the association between neighborhood perceptions (IV) and neighborhood satisfaction (DV).**

| Model Specification | Intercept | Neighborhood Perception (IV) | Personality Trait (M) | Interaction (IV × M) |
|---|---|---|---|---|
| IV = Disorder, M = Openness | 3.90 (0.04)** | -0.50 (0.04)** | -0.08 (0.05) | -0.07 (0.06) |
| IV = Disorder, M = Conscientiousness | 3.91 (0.04)** | -0.47 (0.04)** | 0.20 (0.05)** | -0.02 (0.05) |
| IV = Disorder, M = Extraversion | 3.91 (0.04)** | -0.47 (0.04)** | 0.10 (0.04)* | 0.06 (0.04) |
| IV = Disorder, M = Agreeableness | 3.92 (0.04)** | -0.44 (0.04)** | 0.25 (0.05)** | 0.02 (0.05) |
| IV = Disorder, M = Neuroticism | 3.91 (0.04)** | -0.44 (0.04)** | -0.21 (0.04)** | 0 (0.03) |
| IV = Cohesion, M = Openness | 3.92 (0.05)** | 0.29 (0.04)** | 0.08 (0.05) | 0.01 (0.04) |
| IV = Cohesion, M = Conscientiousness | 3.90 (0.04)** | 0.28 (0.04)** | 0.17 (0.06)** | 0.04 (0.05) |
| IV = Cohesion, M = Extraversion | 3.91 (0.04)** | 0.29 (0.04)** | 0.04 (0.05) | -0.01 (0.04) |
| IV = Cohesion, M = Agreeableness | 3.90 (0.04)** | 0.27 (0.04)** | 0.26 (0.06)** | 0.05 (0.06) |
| IV = Cohesion, M = Neuroticism | 3.91 (0.04)** | 0.27 (0.04)** | -0.26 (0.04)** | 0.03 (0.04) |

**$p < 0.01$.

*$p < 0.05$

**Table 4. Neighborhood perceptions (M) as mediators of the association between personality traits (IV) and neighborhood satisfaction (DV).**

| Personality Trait (IV) | Neighborhood Perception (M) | Direct Effect | | Mediated Effect | |
|---|---|---|---|---|---|
| | | Estimate | *p*-value | Estimate | *p*-value |
| Openness | Disorder | -0.071 | 0.350 | 0.113 | 0.184 |
| Conscientiousness | Disorder | 0.197 | 0.003 | 0.034 | 0.606 |
| Extraversion | Disorder | 0.244 | 0.001 | 0.094 | 0.298 |
| Agreeableness | Disorder | 0.091 | 0.059 | 0.010 | 0.862 |
| Neuroticism | Disorder | -0.215 | < 0.001 | -0.097 | 0.091 |
| Openness | Cohesion | 0.081 | 0.274 | -0.029 | 0.476 |
| Conscientiousness | Cohesion | 0.173 | 0.214 | 0.055 | 0.261 |
| Extraversion | Cohesion | 0.257 | 0.028 | 0.059 | 0.183 |
| Agreeableness | Cohesion | 0.035 | 0.684 | 0.084 | 0.002 |
| Neuroticism | Cohesion | -0.260 | < 0.001 | -0.048 | 0.233 |

Table 4 reports a series of mediated models in which we estimate the mediated effect of a personality trait on neighborhood satisfaction through a perceived neighborhood characteristic, and thus provide an answer to Research Question 3: *is personality's association with satisfaction mediated by perception?* In this table, each row reports the results of one model. For example, the 'IV = Openness, M = Disorder' model reported in the first row reports the results of a mediation model in which the association between openness and neighborhood satisfaction is mediated by perceptions of neighborhood disorder. For each model, we report both the estimated direct effect of a personality trait on neighborhood satisfaction, and the estimated mediated effect of the trait on satisfaction as mediated by a perception. We observe only one statistically significant mediated effect: agreeableness is not directly associated with neighborhood satisfaction (direct effect: $b = 0.035$, $p = 0.684$), but is indirectly associated (mediated effect: $b = 0.084$, $p = 0.002$) via perceptions of greater social cohesion. That is, more agreeable people are more satisfied with their neighborhoods *because* they perceive their neighborhoods to have more social cohesion. These findings offer mixed support for the *rose-colored glasses* model (paths E & B in Fig 1). In general, personality traits do not give people 'rose-colored glasses' that alter how they perceive their neighborhoods in a way that impacts their neighborhood satisfaction. However, agreeableness may be a notable exception: an individual's agreeableness is associated with perceptions of greater social cohesion (i.e., agreeable people view their neighborhood's cohesion through rose-colored glasses), which in turn is associated with greater neighborhood satisfaction.

## Discussion

Urban scholars have theorized several different pathways to and correlates of neighborhood satisfaction, however the potential association between personality and neighborhood satisfaction is less clear. In order to explore possible associations, we tested direct and indirect pathways by which personality might affect neighborhood satisfaction. First, we hypothesized that direct associations exist between Big Five personality traits and neighborhood satisfaction (**Research Question 1**; path A in Fig 1). We found a significant relationship between neighborhood satisfaction and each personality trait except openness. Furthermore, the direction and magnitude of the correlations were comparable to meta-analytic estimates of the relationships between personality and other domains of well-being. These results indicate that personality is associated with neighborhood satisfaction in much the same way as it is associated with other domains of well-being.

Next, we were interested in whether neighborhood satisfaction depends on congruence between individual and environmental characteristics (i.e., the person-environment fit model). Specifically, we explored whether personality traits moderate the association of perceptions of neighborhood cohesion and disorder with neighborhood satisfaction (**Research Question 2**; path D in Fig 1). However, we did not find any support for a potential moderating effect. That is, we find no evidence that perceptions of the neighborhood influence the neighborhood satisfaction of some more than others. For example, contrary to our expectations, conscientious individuals did not experience a stronger negative association between neighborhood disorder and neighborhood satisfaction, and extraverted individuals did not experience a stronger positive association between neighborhood social cohesion and neighborhood satisfaction.

Finally, we were interested in whether personality is associated with neighborhood satisfaction because it shapes how we perceive the world (i.e., the rose-colored glasses model). Specifically, we explored whether the association between personality and neighborhood satisfaction is mediated by perceptions of neighborhood cohesion and disorder (**Research Question 3**; paths E & B in Fig 1). We found only limited support for this indirect association: the association between agreeableness and neighborhood satisfaction is mediated by perceptions of cohesion. That is, agreeableness is associated with perceptions that the neighborhood is more socially cohesive, which in turn are associated with greater neighborhood satisfaction. This effect may be driven by the fact that, among personality traits, agreeableness has the strongest association with personal values and is particularly associated with the values of universalism (i.e., promoting the welfare of all people and nature) and belevolence (i.e., promoting the welfare of people you are close to) [70]. In contrast, contrary to our expectations, a neurotic person's overly negative perception of neighborhood disorder does not lead to a decrease in satisfaction, and an extravert's rosy perception of social cohesion does not lead to an increase in satisfaction. Although only one of the 10 mediated pathways we considered was statistically significant, the significant mediation of agreeableness through perceptions of cohesion suggests that personality can alter perceptions of neighborhoods in a way that is relevant for neighborhood satisfaction, and therefore that the rose-colored glasses model warrants further attention.

The fact that personality is associated with neighborhood satisfaction in the same ways as personality is associated with subjective well-being highlights the potential for drawing on theoretical and empirical developments in positive psychology for understanding neighborhood satisfaction. However, the lack of support for either the person-environment fit or rose-colored glasses models means that the underlying mechanisms for these associations remain unknown. There are at least two possible methodological explanations for the observed lack of support for these models. First, despite our large sample, we may lack the statistical power necessary to detect very small indirect moderated or mediated effects. Although these analyses may be underpowered to detect small effects, the effect sizes we observe are smaller than would be practically meaningful (i.e., to actually matter in a person's daily life). For example, we expected that the negative association between disorder and neighborhood satisfaction ($b = -0.47$) would be more negative for conscientious individuals ($b = -0.02$). Although this tiny estimated indirect effect may have been statistically significant in a larger sample, we believe it is too small to actually matter for individuals' experiences of their neighborhoods. Second, consistent with past research on neighborhood satisfaction [2], we observe a restricted range of neighborhood satisfaction levels, with nearly three-quarters of the sample reporting they were either 'somewhat satisfied' or 'very satisfied' with their neighborhood, but less than 5% reporting they were 'very dissatisfied' with their neighborhood ($M = 3.89$). Restricted ranges are known to downwardly bias correlations and other associations [71], which future

studies may seek to overcome by oversampling respondents who are dissatisfied with their neighborhoods.

There are a number of other plausible mechanisms for the apparent association between personality and neighborhood satisfaction that we were unable to test in these data, but which offer directions for future research. First, this association could be spurious, driven by spatial clustering processes. For example, if personality traits are spatially clustered [50, 72–74], and neighborhood satisfaction is also spatially clustered [2], this could make personality and neighborhood satisfaction appear related even if they are driven by independent spatial clustering processes. Second, the 'environment' which personality is hypothesized to moderate in the person-environment fit model may not be the perceived neighborhood environment, but instead one's own economic situation. For example, while income allows individuals to live in more desirable and satisfying neighborhoods, this income-satisfaction link may be exacerbated by neuroticism [75, 76]. Third, the indirect association of personality with neighborhood satisfaction may be mediated by intervening variables other than neighborhood perceptions, as hypothesized by the rose-colored glasses model. For example, certain personality traits are associated with wealth [77], which in turn may allow individuals to live in more desirable and satisfying neighborhoods. Finally, there is evidence to suggest that personality traits could affect the likelihood of experiencing certain life events in the first place [78]. Some of these life events (e.g., employment and marriage) would almost certainly have downstream consequences for living arrangements and subsequent neighborhood satisfaction, which could further obfuscate relationships between key variables of interest.

While further research is needed on personality and neighborhood satisfaction, this work contributes to the literature in several important ways. First, we offer an integrated model which accounts for multiple potential pathways by which individual differences, perceived and objective neighborhood characteristics might influence neighborhood satisfaction. Second, we conducted one of the first direct investigations of the role of personality in neighborhood satisfaction, and did so with a widely-used and empirically validated personality scale [67]. Third, we found evidence that personality shares a similar relationship with neighborhood satisfaction as it does with other domains of well-being. Fourth, we tested two potential indirect mechanisms for the observed personality-satisfaction association. This contributes to the wider literature on personality and subjective well-being [42, 43], and does so in a way that integrates personality traits and other social-cognitive constructs (i.e., perceived neighborhood characteristics) in understanding neighborhood satisfaction. This integration is analogous to other research that has sought to integrate dispositional traits, characteristic adaptations (i.e., motives, goals, values, etc.), and other critical constructs in the prediction of broad well-being [15, 47, 79]. Although neither of our indirect mechanisms were supported, it highlights the need for future research on understanding why personality is associated with neighborhood satisfaction, perhaps by looking to research in positive psychology on personality and subjective well-being.

There are a few limitations to consider in the present study. First, our findings are based on cross-sectional data, so we cannot establish a causal direction between personality and neighborhood satisfaction. Adult personality traits remain relatively stable at the individual level [80]. Changes that do occur are most often attributable to pivotal life events or purposeful intervention [81, 82]. In the present study, the tendency for personality traits to remain relatively stable means that the most likely causal direction is from personality to neighborhood satisfaction. Second, we used a single-item measure of neighborhood satisfaction. Although this measure of neighborhood satisfaction is widely-used [2] and therefore allows our results to be compared to past studies, it may have lower reliability than a multi-item scale and may have reduced validity if neighborhood satisfaction is in fact a multidimensional construct. Our

single-item measure of cohesion, and two-item measure of disorder, may raise similar reliability concerns, however both variables were associated with neighborhood satisfaction in the expected directions. Third, our data is representative of adults in the state of Michigan and our findings are expected to generalize to this population, which is demographically and politically similar to the United States as a whole. However, because personality is known to exhibit regional variation [50, 72–74], future studies are necessary that examine these relationships in other regions or countries.

This work represents an initial step in the exploration of the direct and indirect associations between personality and neighborhood satisfaction, and therefore lays the groundwork for future research in urban planning to consider the role of personality [9]. Our hypotheses of direct relationships between personality and neighborhood satisfaction were supported, and should inform future research about individual differences and subjective well-being, as well as urban planning efforts to leverage social science for community improvement. We were unable to find evidence for more complex indirect relationships, which given the observed direct relationships remains a promising area for further research. From a planning perspective, a deeper understanding of the role that psychology plays in neighborhood satisfaction could enhance the impact of resources devoted to building more satisfying neighborhoods. Specifically, given the many competing priorities in communities, it will be critical to understand the individual circumstances under which a new amenity or initiative might impact neighborhood satisfaction.

## Author Contributions

**Conceptualization:** Zachary P. Neal, Brian Brutzman.

**Formal analysis:** Zachary P. Neal, Brian Brutzman.

**Methodology:** Zachary P. Neal, Brian Brutzman.

**Supervision:** Zachary P. Neal.

**Writing – original draft:** Zachary P. Neal, Brian Brutzman.

**Writing – review & editing:** Zachary P. Neal, Brian Brutzman.

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
