## [Decision Letter · Decision Letter 0]

18 Nov 2022

PONE-D-22-24984

The role of personality in neighborhood satisfaction

PLOS ONE

Dear Dr. Zachary,

Thank you for submitting your manuscript to PLOS ONE. After careful consideration, we feel that it has merit but does not fully meet PLOS ONE’s publication criteria as it currently stands.

We have received two reviews of your manuscript. Both reviewers highlight some issues that need addressing. Reviewer 1 provides some valuable comments on how the theoretical basis and hypothesis can be improved including recent literature references. Reviewer 2 requests more explanations and justifications of the analytical strategy and interpretation of results.

On the basis of the assessments received, we invite you to submit a revised version of the manuscript that addresses the points raised during the review process.

We look forward to receiving your revised manuscript.

Kind regards,

Manuel Wolff

Academic Editor

PLOS ONE

Journal Requirements:

Reviewers' comments:

Reviewer's Responses to Questions

**Comments to the Author**

1. Is the manuscript technically sound, and do the data support the conclusions?

Reviewer #1: Partly

Reviewer #2: Partly

2. Has the statistical analysis been performed appropriately and rigorously? 

Reviewer #1: Yes

Reviewer #2: No

3. Have the authors made all data underlying the findings in their manuscript fully available?

Reviewer #1: Yes

Reviewer #2: Yes

4. Is the manuscript presented in an intelligible fashion and written in standard English?

Reviewer #1: No

Reviewer #2: No

5. Review Comments to the Author

Reviewer #1: This is a study of high scientific value that deserves to be published.

My congratulations and a cordial greeting to the authors.

However, I noticed some important shortcomings that require improvement.

The following questions should be addressed:

Introduction

The second paragraph in the introduction requires reformulation and supplementation with key literature.

The problem and its importance as well as the aims of the work are not clearly presented (e.g. why it is important to study the direct and indirect role of personality). Additionally, there should be a brief overview of the key literature in the introduction, but there is no reference to the literature on personality and personality-well-being relationships in this paragraph. For example, see:

- Lines (rows) 19-20:

Personality is often conceptualized as consisting of five traits: openness to experience, …

This sentence is unfounded - Please indicate the theory behind such traits and at least one reference to the publication in which they are described.

- Lines (rows) 22-26:

Consistent with prior research on the association between personality and subjective well-being, … [?] Also this sentence is unfounded - Please indicate at least one meta-analysis based on these prior research that can be a base for your hypothesis that there will be association between neighborhood satisfaction and five personality traits.

- Line 28 (and 104):

The person-environment fit model – please indicate a relevant reference (s).

- Line 31 (and 122):

the rose-colored glasses model - please indicate a relevant reference (s).

Background

Pathways to neighborhood satisfaction

Throughout the article, the main variables in the study representing the subjective model (Path B in Figure 1) - the individual perception of neighborhood cohesion and disorder - are merely labeled and they should be clearly defined or explained in this section.

The direct role of personality

Neighbourhood satisfaction is reported in line with the latest publications, but personality theories as well as data on the relationship between personality traits and SWB are relatively old.

Please also provide some up-to-date information as there have been plenty of studies on personality and well-being since 2008 (e.g. Anglim & Grant, 2016; Bojanowska & Urbańska, 2021; Stephan, 2009; Stolarski, 2016; Suldo, Minch, & Hearon, 2015; Szcześniak, Sopińska, & Kroplewski, 2019; Zalewska, 2018), including longitudinal research (e.g. Fetvadjiev & He, 2019) and meta-analyses (Anglim et al, 2020; Steel, Schmidt, Bosco & Uggerslev, 2019).

The indirect roles of personality

It seems, that the broad personality theories including personality traits and characteristic adaptations (e.g. needs, values, attitudes) - such as the Five Factor Theory of Personality-FFT (McCrae & Costa 1999, 2008; McCrae & Suttin, 2018) and the New Big Five theory of personality by McAdams and Pals (2006) - could be used as basis for hypotheses about indirect roles of personality traits (for moderation and especially for mediation analyses) on neighborhood satisfaction, just as they were used for well-being (Zalewska, Nezlek, Zięba, 2018). These theories can also add new context to the Discussion section.

Person(ality)-environment fit.

There are no logical reasons to hypothesize in this section because the perception of neighborhood is not considered in the context of personality traits and their components and there is no information as to how and why perceptions of neighborhood cohesion and disorder are related to personal values, and how values are related with traits.

The rationale given for the hypotheses for question 2 (Path D in Figure 1) is unclear, superficial, and misleading if the reader is familiar with the results of the meta-analysis by Parks-Leduc et al. (2015), which inform that it is not extraversion but agreeableness that is positively related to social values, self-transcendence (benevolence, universalism) and conservation (tradition, conformism and security), and also conscientiousness with conservation. Extraversion is related positively to person-focused values, openness to change (especially stimulation and hedonism) and self-enhancement (power and achievements).

Methods

Measures

The statement “I live in a close-knit neighborhood" does not allow to better understand what means perception of neighborhood social cohesion, so the clear definition of this variable in Background section is very needed.

Results

Table 3 should be supplemented with the bivariate correlations between all the variables analyzed in this study. These are primary data that can be used for secondary data analyzes, such as quantitative literature reviews and meta-analyzes (see: Anglim et al. 2020; Hill and Curran, 2016).

Discussion

In my opinion, the result indicating a significant and specific confirmation that the impact of the Agreeableness on satisfaction with the neighborhood is mediated by the perception of social cohesion is very important and should be more exposed (not neglected).

Also, references to stability and change in personality traits are very old, suggesting that knowledge in pre-2000 literature is universal, although many more recent publications present different knowledge (e.g. Bleidorn, Hopwood, and Lucas, 2018; Costa, McCrae, and Löckenhoff, 2019; Denissen, Luhmann, Chung, Bleidorn, 2018; Roberts, Luo, Briley, Chow, Su and Hill, 2017).

Direct quotations

In the case of direct quotations from other publications, the pages of origin should be given, e.g. “are the genuine locale of the blase attitude" [5, p. ???].

Similar corrections should be made for lines (rows):

13-14, 72, 90-93, 96, 110-111, 127-129, 188-189, 198-201, 218-220.

It would be best to paraphrase sources and better fit material to the context of your work and writing style, and then there is no need to give pages.

References

Most of the references to personality theory, personality traits and their stability (before 2000), and trait-SWB relationships are very old (one from 2015, one from 2008, the rest much older).

More recent references to broad personality theories as well as trait-SWB relationships, especially those based on more current meta-analyses (e.g. Anglim et al., 2020; Steel, Schmidt, Bosco and Uggerslev, 2019) and longitudinal studies (e.g. Fetvadjiev & He, 2019), should be used in the background and in the discussion.

References used in this review:

Anglim, J., & Grant, S. (2016). Predicting Psychological and Subjective Well-Being from Personality: Incremental Prediction from 30 Facets Over the Big 5. Journal of Happiness Studies, 17, 59-80.

Anglim, J., Horwood, S., Smillie, L. D., Marrero, R. J., & Wood, J. K. (2020). Predicting psychological and subjective well-being from personality: A meta-analysis. Psychological Bulletin, 146(4), 279–323. http://doi.org/10.1037/bul0000226

Bleidorn, W., Hopwood, C. J., Lucas, R. E. (2018). Life events and personality trait change. Journal of Personality, 86, 83-96.

Bojanowska, A., & Urbańska, B. (2021). Individual values and well‐being: The moderating role of personality traits. International Journal of Psychology, 56(5), 698-709. http://doi.org/10.1002/ijop.12751

Costa Jr, P. T., McCrae, R. R., & Löckenhoff, C. E. (2019). Personality Across the Life Span. Annual Review of Psychology, 70, 423-448.

Denissen, J.J.A., Luhmann, M. Chung, J.M. Bleidorn, W. (2018). Transactions between life events and personality traits across the adult lifespan. Journal of Personality and Social Psychology. Advance online publication. http://dx.doi.org/10.1037/pspp0000196

Fetvadjiev, V. H., & He, J. (2019). The longitudinal links of personality traits, values, and well-being and self-esteem: A five-wave study of a nationally representative sample. Journal of Personality and Social Psychology, 117(2), 448–464. http://doi.org/10.1037/pspp0000212

Hill, A. P., & Curran, T. (2016). Multidimensional perfectionism and burnout: A meta-analysis. Personality and Social Psychology Review, 20, 269-288.

McAdams, D. P., & Pals, J. L. (2006). A New Big Five: Fundamental principles of a science of personality. American Psychologist, 61, 204-217. doi:10.1037/0003-066X.61.3.204.

McCrae, R. R., & Costa Jr, P. T. (1999). A five-factor theory of personality. In L. A. Pervin, & O. P. John (Eds.), Handbook of personality: Theory and research 2nd Ed., (pp.139–153). New York: Guilford.

McCrae, R. R., & Costa Jr, P. T. (2008). A five-factor theory of personality. In O.P. John, R.W. Robins, & L. A. Pervin (Eds.), Handbook of personality: theory and research (3rd edition). New York, NY: Guilford.

McCrae, R. R., & Sutin, A. R. (2018). A five-factor theory perspective on causal analysis. European Journal of Personality, 32, 151–166. DOI: 10.1002/per.2134

Parks-Leduc, L., Feldman, G., & Bardi, A. (2015). Personality traits and personal values: A meta-analysis. Personality and Social Psychology Review, 19(1), 3-29. doi:10.1177/1088868314538548

Roberts, B. W., Luo, J., Briley, D. A., Chow, P. I., Su, R., & Hill, P. L. (2017). A systematic review of personality trait change through intervention. Psychological Bulletin, 143, 117.

Steel, P., Schmidt, J.A., Bosco, F., & Uggerslev, K. (2019). The effects of personality on job satisfaction and life satisfaction: A meta-analytic investigation accounting for bandwidth–fidelity and commensurability. Human Relations, 72, 217 - 247. doi:10.1177/0018726718771465

Stephan, Y. (2009). Openness to experience and active older adults’ life satisfaction: A trait and facet-level analysis. Personality and Individual Differences, 47, 637-641.

Stolarski, M. (2016). Not restricted by their personality: Balanced Time Perspective moderates well-established relationships between personality traits and well-being. Personality and Individual Differences, 100, 140-144.

Suldo, S. M., R. Minch, D., & Hearon, B. V. (2015). Adolescent Life Satisfaction and Personality Characteristics: Investigating Relationships Using a Five Factor Model. Journal of Happiness Studies, 16, 965-983.

Szcześniak, M., Sopińska, B., & Kroplewski, Z. (2019). Big Five personality traits and life satisfaction: The mediating role of religiosity. Religions, 10, 437.

Zalewska, A. M. (2018). Big-Five and Subjective Well-Being: The mediating role of Individualism or Collectivism beliefs and the moderating role of life period. Polish Psychological Bulletin, 49(2), 166-183. DOI: 10.24425/119484.

Zalewska, A. M., Nezlek, J., & Zięba, M. (2018). Integrated approach to personality and well-being. Polish Psychological Bulletin, 49(2), 128–130. DOI: 10.24425/119479.

Reviewer #2: The manuscript draft covers an important issue that is linked to urban science as an interdisciplinary research subject.

The methods section contains a short description of the data acquisition, with a brief remark on the timing of the data collection, followed by a description of the operationalization of the measures and an analysis plan indicating the statistical methods used. The R-code, including the data repository for the statistical analyses is made available via hyperlink. The methods section is comparably short and could benefit from more detailed information regarding operationalization (see below).

Despite the inconclusive results for the role of mediation this study supplies findings that can be useful for urban planning practice as well as future research.

The reviewer recommends publication after the issues listed below have been considered.

Minor issues:

As this paper will be interesting for an interdisciplinary audience, the reviewer suggests including a paragraph, that explains the difference between the concepts moderator and mediator and their operationalization in research. (This concerns Review Question 1.)

Data Section (lines 243-255): It should be clarified, how the data is collected in the SOSS, i.e. that researchers add their questions to the SOSS´s set of core questions. Were the questions about neighborhood satisfaction, personality and perceived neighborhood characteristics part of core questions or added? (This concerns Review Question 1.)

Line 288: Is the Pearson correlation coefficient (parametric) the adequate choice for processing ordinal-scaled data? - Were alternatives (non-parametric) measures of association considered? (this concerns Review Question 2.)

Line 291: With the scale of the dependent variable being ordinal, why were linear regressions chosen? Were other regression models considered?

For the calculated linear regressions, are the assumptions - linear relationship between each predictor variable and the response variable, no high correlation between predictor variables, independence of observations, homoscedasticity, normal distribution of residuals - satisfied? (This concerns Review Question 2.)

Lines 293-296: The causal mediation analysis requires that the sequential ignorability assumption is not violated, therefore a sensitivity analysis, testing for unobserved pre-treatment covariates, is recommended. In connection with the point made in the discussion part, for example alternative mediators to neighborhood perceptions (lines 428-432) would justify a sensitivity analysis. Please explain why a sensitivity analysis is not documented. (This concerns Review Question 2.)

Line 319: the word “this” is used twice. (This concerns Review Question 4.)

Lines 322-324: Can the correlation coefficients: r=0,307 and r=-0,464 really be interpreted as strong? With r=1 and r=-1 representing perfectly positive and negative correlations respectively, why is strong association interpreted at values between 0 and 0.5 for the positive and 0 and -0.5 for the negative correlation respectively? Please explain. (This concerns Review Question 1.)

Lines 408,409: The sentence seems unclear, as it mixes statistical and practical significance without properly explaining what the latter means. (This concerns Review Question 4.)

6. PLOS authors have the option to publish the peer review history of their article (what does this mean?). If published, this will include your full peer review and any attached files.

Reviewer #1: **Yes: **Anna M. Zalewska

Reviewer #2: No

---

## [Decision Letter · Decision Letter 1]

15 Feb 2023

The role of personality in neighborhood satisfaction

PONE-D-22-24984R1

Dear Dr. Zachary,

We’re pleased to inform you that your manuscript has been judged scientifically suitable for publication and will be formally accepted for publication once it meets all outstanding technical requirements.

Kind regards,

Manuel Wolff

Academic Editor

PLOS ONE

Additional Editor Comments (optional):

Reviewers' comments:

Reviewer's Responses to Questions

**Comments to the Author**

1. If the authors have adequately addressed your comments raised in a previous round of review and you feel that this manuscript is now acceptable for publication, you may indicate that here to bypass the “Comments to the Author” section, enter your conflict of interest statement in the “Confidential to Editor” section, and submit your "Accept" recommendation.

Reviewer #1: All comments have been addressed

Reviewer #2: All comments have been addressed

2. Is the manuscript technically sound, and do the data support the conclusions?

Reviewer #1: Yes

Reviewer #2: Yes

3. Has the statistical analysis been performed appropriately and rigorously? 

Reviewer #1: Yes

Reviewer #2: Yes

4. Have the authors made all data underlying the findings in their manuscript fully available?

Reviewer #1: Yes

Reviewer #2: Yes

5. Is the manuscript presented in an intelligible fashion and written in standard English?

Reviewer #1: Yes

Reviewer #2: Yes

6. Review Comments to the Author

Reviewer #1: Thank you very much for the revised version of the manuscript.

The authors took into account all my suggestions and comments in this version of the article. I am glad with the substantive amendments and I greatly appreciate this work.

I congratulate the authors for the excellent ideas, interesting research, and a very well-written article.

Reviewer #2: In line 164 of the revised manuscript (R1): the abbreviation is SWL it should be SWJ, shouldn´t it?

In line 231 there seems to be one " too much.

7. PLOS authors have the option to publish the peer review history of their article (what does this mean?). If published, this will include your full peer review and any attached files.

Reviewer #1: **Yes: **Anna M. Zalewska

Reviewer #2: No

---

## [Editor Report · Acceptance letter]

17 Feb 2023

PONE-D-22-24984R1 

The role of personality in neighborhood satisfaction  

Dear Dr. Neal:

I'm pleased to inform you that your manuscript has been deemed suitable for publication in PLOS ONE. Congratulations! Your manuscript is now with our production department. 

Kind regards, 

on behalf of

Dr. Manuel Wolff 

Academic Editor

PLOS ONE